# Reporting guidelines in medical artificial intelligence: a systematic review and meta-analysis

Fiona R. Kolbinger [1,2,3,4,5,6,11], Gregory P. Veldhuizen[1,11], Jiefu Zhu[1], Daniel Truhn [7] & Jakob Nikolas Kather [1,8,9,10] ✉

## Abstract

**Background** The field of Artificial Intelligence (AI) holds transformative potential in medicine. However, the lack of universal reporting guidelines poses challenges in ensuring the validity and reproducibility of published research studies in this field.

**Methods** Based on a systematic review of academic publications and reporting standards demanded by both international consortia and regulatory stakeholders as well as leading journals in the fields of medicine and medical informatics, 26 reporting guidelines published between 2009 and 2023 were included in this analysis. Guidelines were stratified by breadth (general or specific to medical fields), underlying consensus quality, and target research phase (preclinical, translational, clinical) and subsequently analyzed regarding the overlap and variations in guideline items.

**Results** AI reporting guidelines for medical research vary with respect to the quality of the underlying consensus process, breadth, and target research phase. Some guideline items such as reporting of study design and model performance recur across guidelines, whereas other items are specific to particular fields and research stages.

**Conclusions** Our analysis highlights the importance of reporting guidelines in clinical AI research and underscores the need for common standards that address the identified variations and gaps in current guidelines. Overall, this comprehensive overview could help researchers and public stakeholders reinforce quality standards for increased reliability, reproducibility, clinical validity, and public trust in AI research in healthcare. This could facilitate the safe, effective, and ethical translation of AI methods into clinical applications that will ultimately improve patient outcomes.

## Plain Language Summary

Artificial Intelligence (AI) refers to computer systems that can perform tasks that normally require human intelligence, like recognizing patterns or making decisions. AI has the potential to transform healthcare, but research on AI in medicine needs clear rules so caregivers and patients can trust it. This study reviews and compares 26 existing guidelines for reporting on AI in medicine. The key differences between these guidelines are their target areas (medicine in general or specific medical fields), the ways they were created, and the research stages they address. While some key items like describing the AI model recurred across guidelines, others were specific to the research area. The analysis shows gaps and variations in current guidelines. Overall, transparent reporting is important, so AI research is reliable, reproducible, trustworthy, and safe for patients. This systematic review of guidelines aims to increase the transparency of AI research, supporting an ethical and safe progression of AI from research into clinical practice.

[1]Else Kroener Fresenius Center for Digital Health, TUD Dresden University of Technology, Dresden, Germany. [2]Department of Visceral, Thoracic and Vascular Surgery, University Hospital and Faculty of Medicine Carl Gustav Carus, TUD Dresden University of Technology, Dresden, Germany. [3]Weldon School of Biomedical Engineering, Purdue University, West Lafayette, IN, USA. [4]Regenstrief Center for Healthcare Engineering, Purdue University, West Lafayette, IN, USA. [5]Department of Biostatistics and Health Data Science, Richard M. Fairbanks School of Public Health, Indiana University, Indianapolis, IN, USA. [6]Indiana University Simon Comprehensive Cancer Center, Indiana University School of Medicine, Indianapolis, IN, USA. [7]Department of Diagnostic and Interventional Radiology, University Hospital Aachen, Aachen, Germany. [8]Department of Medicine III, University Hospital RWTH Aachen, Aachen, Germany. [9]Department of Medicine I, University Hospital Dresden, Dresden, Germany. [10]Medical Oncology, National Center for Tumor Diseases (NCT), University Hospital Heidelberg, Heidelberg, Germany. [11]These authors contributed equally: Fiona R. Kolbinger, Gregory P. Veldhuizen. ✉e-mail: jakob-nikolas.kather@alumni.dkfz.de

The field of Artificial Intelligence (AI) is rapidly growing and its applications in the medical field have the potential to revolutionize the way diseases are diagnosed and treated. Despite the field still being in its relative infancy, deep learning algorithms have already proven to perform at parity with or better than current gold standards for a variety of tasks related to patient care. For example, deep learning models perform on par with human experts in classification of skin cancer[1], aid in both the timely identification of patients with sepsis[2] and respective adaptation of the treatment strategy[3], and can identify genetic alterations from histopathological imaging across different cancer types[4]. Due to the black box nature of many AI-based investigations, it is critical that the methodology and results of the findings are reported in a thorough, transparent and reproducible manner. However, despite this need, such measures are often omitted[5]. High reporting standards are vital in ensuring that public trust, medical efficacy and scientific integrity are not compromised by erroneous, often overly positive performance metrics due to flaws such as skewed data selection or methodological errors such as data leakage.

To address these challenges, numerous reporting guidelines have been developed to regulate AI-related research in preclinical, translational, and clinical settings. A reporting guideline is a set of criteria and recommendations designed to standardize the reporting of research methodologies and findings. These guidelines aim to ensure the inclusion of minimum essential information within research studies and thereby enhance transparency, reproducibility, and the overall quality of research reporting[6,7]. While clinical treatment guidelines typically describe a summary of standards of care based on existing medical evidence, there is no universal standard approach for the development of reporting guidelines regarding what information should be provided when attempting to publish findings from a scientific investigation. Consequently, the quality of reporting guidelines can vary depending on the methods used to reach consensus as well as the individuals involved in the process. The Delphi method, when employed by a panel of authoritative experts in the relevant field, is generally considered to be the most appropriate means of obtaining high-quality agreement[8]. This method describes a structured technique in which experts cycle through several rounds of questionnaires, with each round resulting in an updated questionnaire that is provided to participants along with a summary of responses in the subsequent iteration. This pattern is repeated until consensus is reached.

Another factor to consider when developing reporting guidelines for medical AI is their scope. Reporting guidelines may be specific to the unique needs of a single clinical specialty or intended to be more general in nature. In addition, due to the highly dynamic nature of AI research, these guidelines require frequent reassessment to safeguard against obsolescence. As a consequence of the breadth of stakeholders involved in the development and regulation of medical AI, including government organizations, academic institutions, publishers and corporations, a multitude of reporting guidelines have arisen. The repercussion of this is a notable lack of clarity for researchers as to which guidelines to follow, uncertainty whether or not guidelines exist for their specific domain of research, and whether or not reporting standards can be expected to be enforced by publishers of mainstream academic journals. As a result, despite the abundance of reporting guidelines for healthcare, only a fraction of research items adheres to them[9–11]. This reflects a deficiency on the part of researchers and scholarly publishers alike.

This systematic review provides an overview of existing reporting guidelines for AI-related research in medicine that have been published by research consortia, federal institutions, or adopted by medical and medical informatics publishers. It summarizes the key elements that are near-universally considered necessary when reporting findings to ensure maximum reproducibility and clinical validity. These key elements include descriptions of the clinical rationale, the data that reported models are based on, and of the training and validation process. By highlighting guideline items that are widely agreed upon, our work aims to provide orientation to researchers, policymakers, and stakeholders in the field of medical AI and form a basis for the development of future reporting guidelines with the goal of ensuring maximum reproducibility and clinical translatability of AI-related medical research. In addition, our summary of key reporting items may provide guidance for researchers in situations where no high-quality reporting guideline currently exists for the topic of their research.

## Methods
### Search strategy
We report the results of this systematic review following the PRISMA 2020 statement for reporting systematic reviews[12]. To cover the breadth of published AI-related reporting guidelines in medicine, our search strategies included three sources: (i) Guidelines published as scholarly research publications listed in the database PubMed and in the EQUATOR Network's library of reporting guidelines (https://www.equator-network.org/library/), (ii) AI-related statements and requirements of international federal health agencies, and (iii) relevant journals in Medicine and Medical Informatics. The search strategy was developed by three authors with experience in medical AI research (FRK, GPV, JNK), and no preprint servers were included in the search.

PubMed was searched on June 26, 2022, without language restrictions, for literature published since database inception, on AI guidelines in the fields of preclinical, translational, and clinical medicine, using the keywords ("Artificial Intelligence" OR "Machine Learning" OR "Deep Learning") AND ("consensus statement" OR "guideline" OR "checklist"). The EQUATOR Network's library of reporting guidelines was searched on November 14, 2023, using the keywords "Artificial Intelligence", "Machine Learning" and "Deep Learning". Additionally, statements and requirements of the federal health agencies of the United States (Food and Drug Administration, FDA), the European Union (European Medicines Agency, EMA), the United Kingdom (Medicines and Healthcare Products Regulatory Agency), China (National Medical Products Association), and Japan (Pharmaceuticals and Medical Devices Agency) were reviewed with respect to further guidelines and requirements. Finally, the ten journals in Medicine and Medical Informatics with the highest journal impact factors in 2021 according to the Clarivate Journal Citation reports were screened for specific AI/ML checklist requirements for submitted articles. Studies identified as incidental findings were added independent of the aforementioned search process, thereby including studies published after the initial search on June 26, 2022.

### Study selection
Duplicate studies were removed. All search results were independently screened by two physicians with experience in clinical AI research (FRK and GPV) using Rayyan[13]. Screening results were blinded until completion of each reviewer's individual screening. The inclusion criteria were (1) the topic of the publication being AI in medicine and (2) the guideline recommendations being specific to the application of AI methods for either preclinical, translational, or clinical scenarios. Publications were excluded on the basis of (1) not providing actionable reporting guidance, (2) collecting or reassembling guideline items from existing guidelines rather than providing new guideline items or (3) reporting the intention to develop a new, as yet unpublished guideline rather than the guideline itself. Disagreements regarding guideline selection were resolved by judgment of a third reviewer (JNK).

### Data extraction and analysis
Two physicians with experience in clinical AI research (FRK, GPV) reviewed all selected guidelines and extracted the year of publication, the target research phase (preclinical, translational and/or clinical research), the breadth of the guideline (general or specific to a medical subspecialty) and the consensus process as a way to assess the risk of bias. The target research phase was considered preclinical if the guideline regulates theoretical studies not involving clinical outcome data but potentially retrospectively involving patient data, translational if the guideline targets

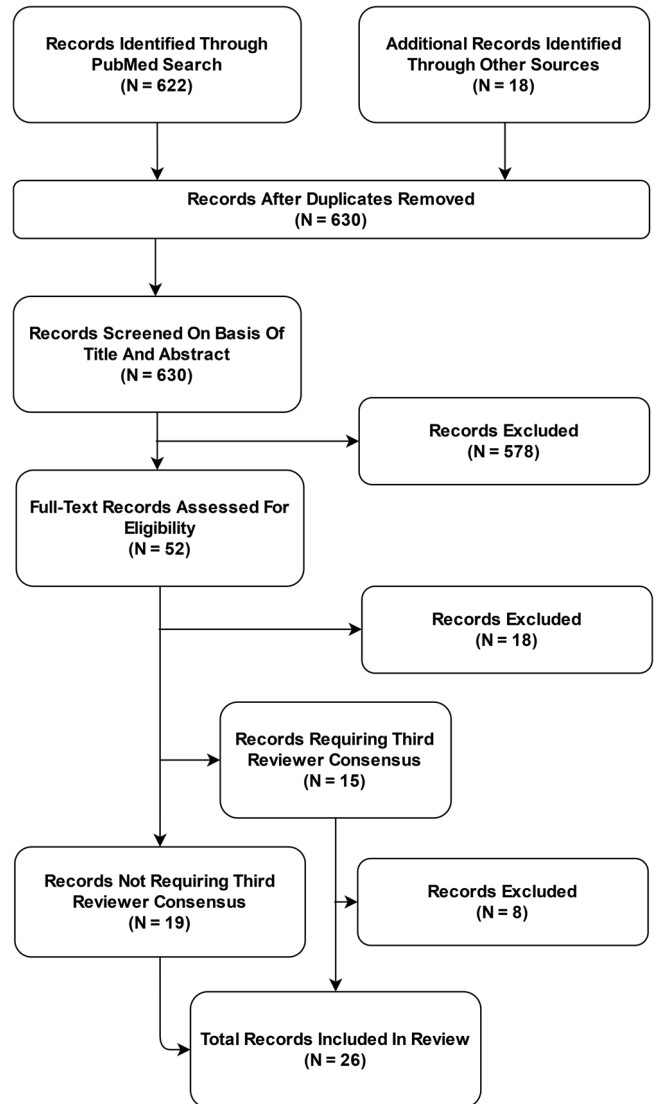

**Fig. 1 | Flowchart of the systematic review and meta-analysis according to the PRISMA 2020 statement for reporting systematic reviews.** Based on a systematic review of academic publications and reporting standards demanded by international federal health institutions and leading journals in the fields of medicine and medical informatics, 26 reporting guidelines published between 2009 and 2023 were included in this analysis.

retrospective or prospective observational trials involving patient data with a potential clinical implication, and clinical if the guideline regulates interventional trials in a clinical setting. The breadth of a guideline was considered general or subject-specific depending on target research areas mentioned in the guideline. Additionally, reporting guidelines were independently graded by FRK and GPV (with arbitration by a third rater, JNK, in case of disagreement) as being either "comprehensive", "collaborative" or "expert-led" in their consensus process. The consensus process of a guideline was classified as expert-led if the method by which it was developed did not appear to be through a consensus-based procedure, if the guideline did not involve relevant stakeholders, or if the described development procedure was not clearly outlined. Guidelines were classified as collaborative if the authors (presumably) used a formal consensus procedure involving multiple experts, but provided no details on the exact protocol or methodological structure. Comprehensive guidelines outlined a structured, consensus-based, methodical development approach involving multiple experts and relevant stakeholders with details on the exact protocol (e.g., using the Delphi procedure).

FRK and GPV extracted each guideline's recommended items for the purpose of creating an omnibus list of all as-yet published guideline items (Supplementary Table 1). FRK and GPV independently evaluated each guideline for the purpose of determining which items from the omnibus list were either fully, partially, or not covered by each publication individually. Aspects that were directly described in a guideline including some details or examples were considered "fully" covered, aspects mentioned implicitly using general terms were considered "partially" covered. Disagreements were resolved by judgment of a third reviewer (JNK). Overlap of guideline content was visualized using pyCirclize[14]. Items recommended by at least 50% of all reporting guidelines or 50% of reporting guidelines with a specified systematic development process (i.e., comprehensive consensus) were considered universal recommendations for clinical AI research reporting.

### Study registration
This systematic review was registered at OSF https://doi.org/10.17605/OSF.IO/YZE6J on August 25, 2023. The protocol was not amended or changed.

### Reporting summary
Further information on research design is available in the Nature Portfolio Reporting Summary linked to this article.

## Results
### Search results
The PubMed database search yielded 622 unique publications; another 18 guidelines were identified through other sources: 8 guidelines were identified through a search of the EQUATOR Network's library of reporting guidelines, two guidelines were identified through review of recommendations of federal agencies; one additional guideline was included based on review of journal recommendations. Another seven additional guidelines were added as incidental findings.

After removal of duplicates, 630 publications were subjected to the screening process. Out of these, 578 records were excluded based on Title and Abstract. Of the remaining 52 full-text articles assessed for eligibility, 26 records were excluded and 26 reporting guidelines were included in the systematic review and meta-analysis (Fig. 1). Interrater agreement for study selection on the basis of full-text records was 71% ($n = 15$ requiring third reviewer out of $n = 52$).

### The landscape of reporting guidelines in clinical AI
A total of 26 reporting guidelines was included in this systematic review. We identified nine comprehensive, six collaborative and eleven expert-led reporting guidelines. Approximately half of all reporting guidelines ($n = 14$, 54%) provided general guidelines for AI-related research in medicine. The remaining publications ($n = 12$, 46%) were developed to regulate the reporting of AI-related research within a specific field of medicine. These included medical physics, dermatology, cancer diagnostics, nuclear medicine, medical imaging, cardiovascular imaging, neuroradiology, psychiatry, and dental research (Table 1, Figs. 2 and 3).

We systematically categorized the reporting guidelines by the research phase that they were aimed at as well as the level of consensus used in their development (Fig. 2, Fig. 3). The majority of guidelines ($n = 20$, 77%) concern AI applications for preclinical and translational research rather than clinical trials. Of these preclinical and translational reporting guidelines, many ($n = 12$) are specific for individual fields of medicine such as cardiovascular imaging, psychiatry or dermatology rather than generally applicable recommendations. In addition, these guidelines tend to more often be expert-led or collaborative ($n = 15$) in nature rather than comprehensive ($n = 5$). This is in contrast to the considerably fewer clinical reporting guidelines ($n = 6$) that are universally general in nature and overwhelmingly comprehensive in their consensus process ($n = 4$). There has been a notable increase in the publication of reporting guidelines in recent years, with 81% ($n = 21$) of included guidelines having been published in or after 2020.

**Table 1 | Summary of reporting guidelines included in this systematic review**

| Guideline | Target research phase | Target study type | Year | Inclusion process | Breadth | Level of consensus |
|---|---|---|---|---|---|---|
| STARE-HI[32] | Clinical | Not specified | 2009 | + | General | Comprehensive |
| Vlhinen[33] | Preclinical | Genomic variant interpretation | 2012 | * | General | Expert-led |
| TRIPOD[34] | Preclinical, Translational | Diagnostic and prognostic predictive modeling | 2015 | % | General | Comprehensive |
| Luo et al.[35] | Preclinical, Translational | Diagnostic and prognostic predictive modeling | 2016 | * | General | Comprehensive |
| Good ML Practice[36] | Not specified | Not specified | 2019 | + | General | Collaborative |
| CLAIM[37] | Preclinical, Translational | Diagnostic and prognostic predictive modeling | 2020 | # | Specific (medical imaging) | Expert-led |
| CONSORT-AI[38] | Clinical | Randomized controlled trials | 2020 | * | General | Comprehensive |
| MI-CLAIM[39] | Preclinical, Translational | Not specified | 2020 | * | General | Collaborative |
| PRIME[40] | Preclinical, Translational | Not specified | 2020 | * | Specific (cardiovascular imaging) | Collaborative |
| SPIRIT-AI[41] | Clinical | Clinical trial protocols for randomized controlled trials | 2020 | * | General | Comprehensive |
| MINIMAR[42] | Preclinical, Translational | Not specified | 2020 | * | General | Expert-led |
| Stevens et al.[43] | Preclinical, Translational | Not specified | 2020 | * | General | Expert-led |
| DOME[44] | Preclinical | Supervised learning on biological data | 2021 | * | General | Collaborative |
| CAIR[45] | Preclinical, Translational | Not specified | 2021 | * | General | Expert-led |
| PIECES[46] | Translational | External validation of diagnostic deep learning systems | 2021 | # | Specific (cancer diagnostics) | Expert-led |
| El Naqa et al.[47] | Preclinical | Research of AI/ML in the field of medical physics | 2021 | # | Specific (medical physics) | Expert-led |
| Zukotynski et al.[48] | Preclinical, Translational | Not specified | 2021 | * | Specific (nuclear medicine) | Expert-led |
| Schwendicke et al.[49] | Preclinical, Translational | Studies on AI in dentistry | 2021 | * | Specific (dental research) | Comprehensive |
| CLEAR Derm[50] | Preclinical, Translational | Dermatological image analysis | 2022 | * | Specific (dermatology) | Comprehensive |
| DECIDE-AI[51] | Translational, Clinical | Clinical decision support systems | 2022 | * | General | Comprehensive |
| Jones et al.[52] | Preclinical, Translational | Skin cancer diagnosis | 2022 | * | Specific (dermatology) | Expert-led |
| R-AI-DIOLOGY[53] | Preclinical, Translational | Not specified | 2022 | * | Specific (neuroradiology) | Expert-led |
| Shen et al.[54] | Preclinical, Translational | Ethics in deep phenotyping | 2022 | * | Specific (psychiatry) | Collaborative |
| Volovici et al.[55] | Translational, Clinical | Avoiding misuse of AI in clinical research | 2022 | # | General | Expert-led |
| Hatt et al.[56] | Preclinical, Translational | Radiomics analyses | 2023 | # | Specific (nuclear medicine) | Collaborative |
| CLEAR[57] | Preclinical, Translational | Radiomics analyses | 2023 | * | Specific (medical imaging) | Comprehensive |

Inclusion process: *: Guideline identified via a systematic, blinded review of the literature (PubMed, EQUATOR Network library of reporting guidelines); +: Guideline identified by additional pre-specified inclusion procedure; #: Guideline added after incidental finding; %: Guideline on reporting of predictive models with AI-specific guideline under development. Target trial phase: Preclinical: Theoretical studies not involving clinical outcome data but potentially retrospectively involving patient data; Translational: Retrospective or prospective observational trials involving patient data with a potential clinical implication; Clinical: Interventional trial in a clinical setting. Level of evidence: Expert-led: No formal procedure; Collaborative: (Presumably) formal consensus procedure involving multiple experts, but no details on exact protocol; Comprehensive: Formal consensus procedure involving multiple experts with details on exact protocol.

**Fig. 2 | Overlap between reporting guidelines and their applicability for various research phases.**
Preclinical guidelines regulate theoretical studies not involving clinical outcome data but potentially retrospectively involving patient data. Translational guidelines target retrospective or prospective observational trials involving patient data with a potential clinical implication. Clinical guidelines regulate interventional trials in a clinical setting. Reporting guidelines catering towards specific research phases are able to be more specific in their items, while those aimed at overlapping research phases tend to necessitate more general reporting items.

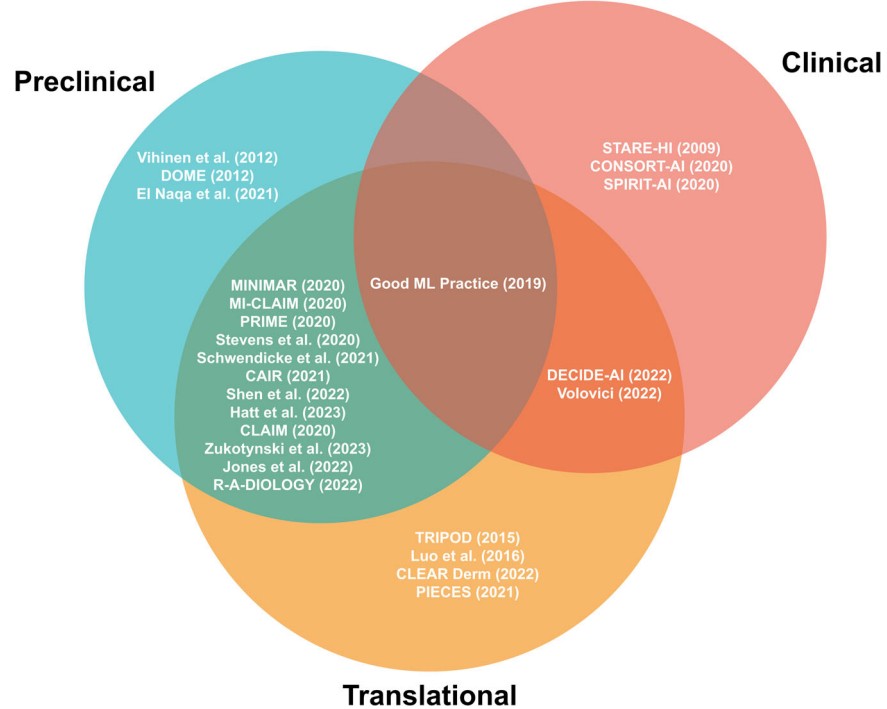

## Consensus in guideline items

The identified guidelines were analyzed with respect to their overlap in individual guideline recommendations (Supplementary Table 1, Fig. 4a, b). A total of 37 unique guideline items were identified. These concerned Clinical Rationale (7 items), Data (11 items), Model Training and Validation (9 items), Critical Appraisal (3 items), and Ethics and Reproducibility (7 items). We were unable to identify a clear weighting towards certain items over others within our primary means of clustering reporting guidelines, namely the consensus procedure and whether the guideline is directed at specific research fields or provides general guidance (Fig. 4b).

Figure 5 summarizes items that were recommended by at least 50% of all guidelines or 50% of guidelines with a specified systematic development process (comprehensive guidelines). These items are considered universal components of studies on predictive clinical AI models.

## Discussion

With the increasing availability of computational resources and methodological advances, the field of AI-based medical applications has experienced significant growth over the last decade. To ensure reproducibility, responsible use and clinical validity of such applications, numerous guidelines have been published, with varying development strategies, structures, application targets, content and support from research communities. We conducted a systematic review of existing guidelines for AI applications in medicine, with a focus on assessing their quality, application areas, and content.

Our analysis suggests that the majority of AI-related reporting guidelines has been conceived by individual (groups of) stakeholders without a formal consensus process and that most reporting guidelines address preclinical and translational research rather than the clinical validation of AI-based applications. Guidelines targeting specific medical fields often result from less rigorous consensus processes than broader guidelines targeting medical AI in general, resulting in some use cases for which several high-evidence guidelines are available (i.e., dermatology, medical imaging), whereas no specialty-independent guideline developed in a formal consensus process is currently available for preclinical research.

Differences in data types and tasks that AI can address in different medical specialties represent a key challenge for the development of guidelines for AI applications in medicine. Many predominantly diagnostics-based specialties such as pathology or radiology rely heavily on different types of imaging with distinct peculiarities and challenges. The need to account for such differences is stronger in preclinical and translational steps of development as compared to clinical evaluation, where AI applications are tested for validity.

Most specialty-specific guidelines address preclinical phases, and these guidelines have predominantly been conceived in less rigorous consensus processes. While individual peculiarities of specific use cases may be clearer in specific guidelines than in more general guidelines, it is conceivable that subject-specific guidelines could result in many guidelines on the same topic when use cases and guideline requirements are similar across fields. To address this issue, stratification by data type could be a potential solution to ensure that guidelines are universal yet specific enough to regulate.

Incorporation of innovations in guidelines represents another challenge, as guidelines have traditionally been distributed in the form of academic publications. In this context, the fact that AI represents a major methodological innovation has been acknowledged by regulating institutions such as the EQUATOR network, which has issued AI-specific counterparts for existing guidelines, including CONSORT(-AI) regulating randomized controlled clinical trials and SPIRIT(-AI) regulating interventional clinical trial protocols. Several other comprehensive high-quality AI-specific guideline extensions are expected to become publicly available in the near future including STARD-AI[15], TRIPOD-AI[16], and PRISMA-AI[17]. Ideally, guidelines should be adaptive and interactive to dynamically integrate new innovations as they emerge. Two quality assessment tools, PROBAST-AI[16] (for risk of bias and applicability assessment of prediction model studies) and QUADAS-AI[18] (for quality assessment of AI-centered diagnostic accuracy studies), will be developed alongside the anticipated AI-specific reporting guidelines.

To prevent the previously mentioned creation of multiple guidelines on the same topic, guidelines could potentially be continuously updated. However, this requires careful management to ensure that guidelines remain relevant and up-to-date without becoming overwhelming or contradictory. On a similar line, it may be worth considering whether AI-specific guidelines should repeat non-AI-specific items, such as ethics statements or Institutional Review Board (IRB) requirements. It may be useful to compare these needs with good scientific practice, to refer to

**Fig. 3 | Existing guidelines on AI in medicine by research phase and level of consensus.** Preclinical guidelines regulate theoretical studies not involving clinical outcome data but potentially retrospectively involving patient data. Translational guidelines target retrospective or prospective observational trials involving patient data with a potential clinical implication. Clinical guidelines regulate interventional trials in a clinical setting. The breadth of guidelines is classified as general or subject-specific depending on target research areas mentioned in the guideline. In terms of the consensus process, comprehensive guidelines are based on a structured, consensus-based, methodical development approach involving multiple experts and relevant stakeholders with details on the exact protocol. Collaborative guidelines are (presumably) developed using a formal consensus procedure involving multiple experts, but provide no details on the exact protocol or methodological structure. Expert-led guidelines are not developed through a consensus-based procedure, do not involve relevant stakeholders, or do not clearly describe the development procedure.

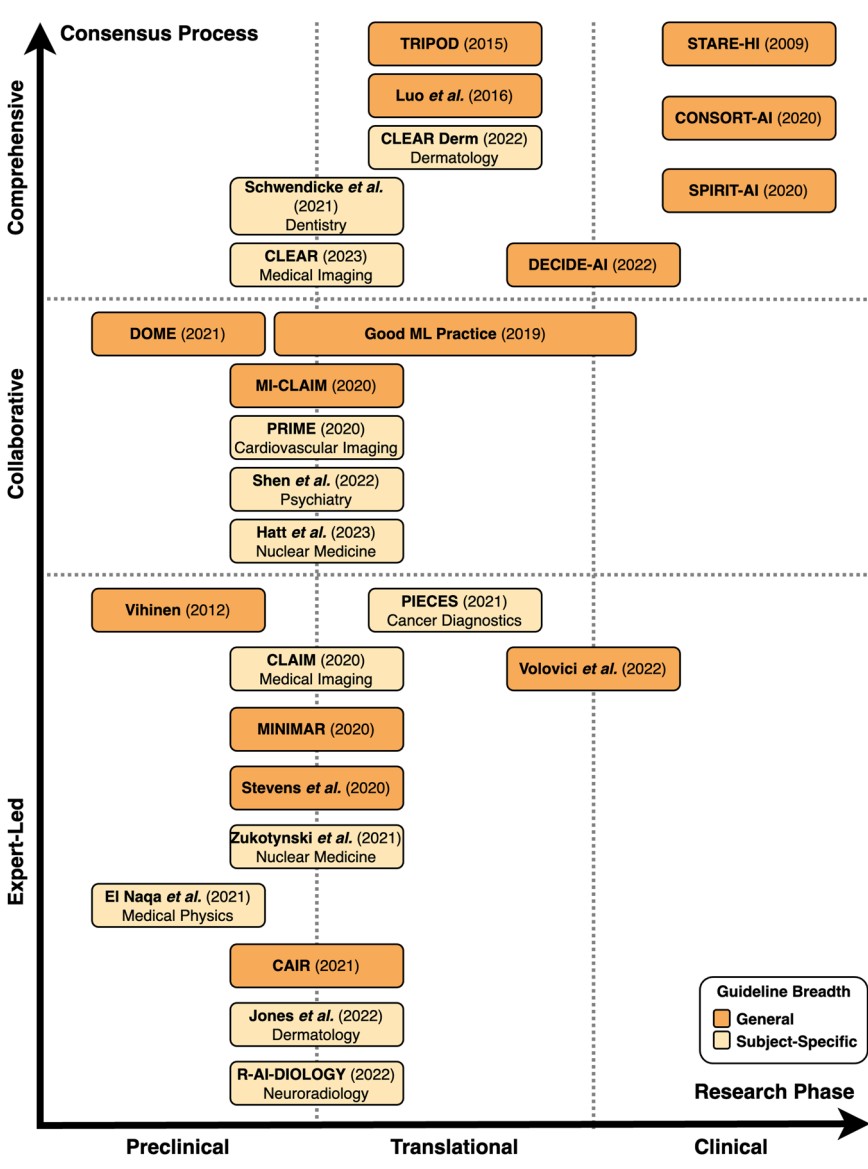

existing resources, and to consider how best to balance comprehensiveness with clarity and ease of use. Whenever new guidelines are being developed, it is advisable to follow available guidance to ensure high guideline quality through methods like a structured literature review and a multi-stage Delphi process[19,20].

Before entering clinical practice, medical innovations must undergo a rigorous evaluation process, and regulatory needs play a crucial role in this process. However, this can lead to undynamic processes, resulting in a gap between large amounts of preclinical research that largely do not enter steps towards clinical translation. Therefore, future guidelines should include items relevant to translational processes, such as regulatory sciences, access, updates, and assessment of feasibility for implementation into clinical practice. Less than half of the guidelines included in this review mentioned such items. By including such statements, better selection of disruptive and clinically impactful research could be made.

Despite various available guidelines, some use cases including preclinical research remain poorly regulated, and it is necessary to address gaps in existing guidelines. For such cases, it is advisable to identify the most relevant general guideline and adhere to key guideline items that are universally accepted and should be part of any AI research in the medical field. As a consequence, researchers can be guided on what to include in their research, and regulatory bodies can be more stringent in

demanding adherence to guidelines. In this context, our review resulted in the finding that many high-impact medical and medical informatics journals do not demand adherence to any guidelines. While peer reviewers can encourage respective additions, more stringency in adherence to guidelines would help ensure the responsible use of AI-based medical applications.

While the content of reporting guidelines in medical AI has been critically reviewed previously[21,22], this is, to our knowledge, the first systematic review on reporting guidelines used in various stages of AI-related medical research. Importantly, this review focuses on guidelines for AI applications in healthcare and intentionally does not consider guidelines for prediction models in general; this has been done elsewhere[10].

The limitations of this systematic review are primarily related to its methodology: First, our search strategy was developed by three of the authors (FRK, GPV, JNK), without any external review of the search strategy[23] and without input from a librarian. Similarly, our systematic search was limited to the publication database PubMed, the EQUATOR Network's library of reporting guidelines (https://www.equator-network.org/library), journal guidelines and guidelines of major federal institutions. An involvement of internal peer reviewers with journalogical experience in the development of the search strategy and an inclusion of preprint servers in the search may have revealed additional guidelines to

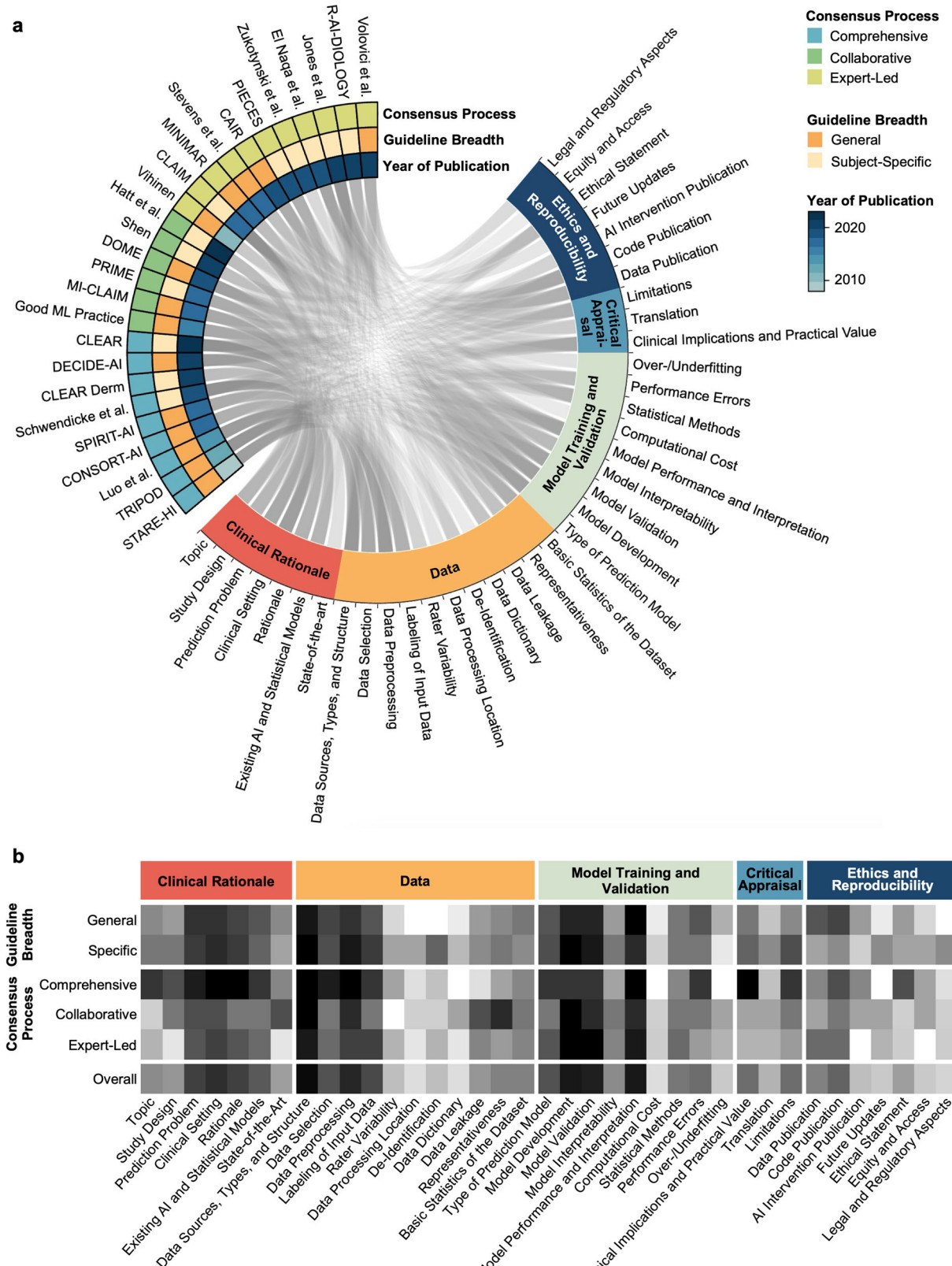

**Fig. 4 | Concordance of medical AI reporting guideline items.** The Circos plot (**a**) displays represented content as a connecting line between guideline and guideline items. The heatmap (**b**) displays the differential representation of specific guideline aspects depending on guideline quality and breadth. Darker color represents a higher proportion of representation of the respective guideline aspect in the respective group of reporting guidelines for medical AI.

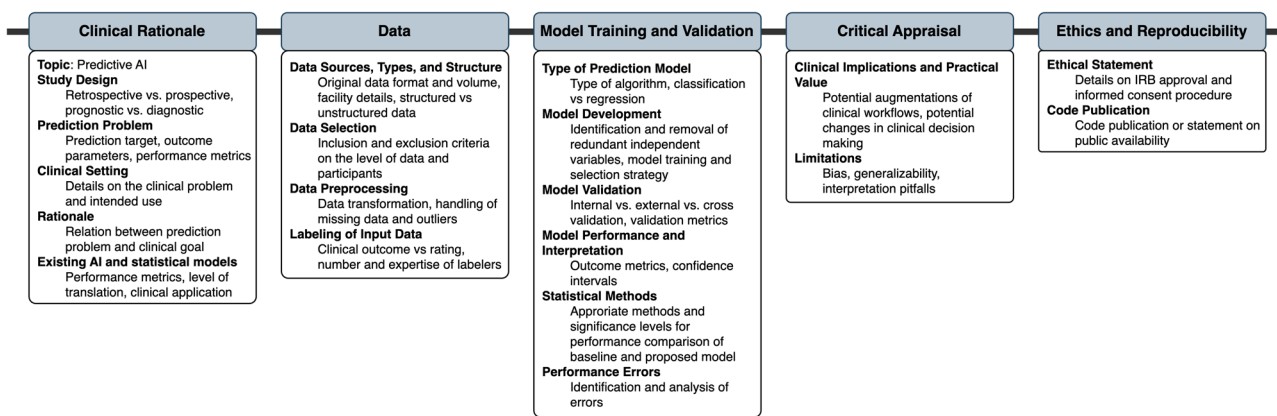

**Fig. 5 | Universal components of studies on predictive clinical AI models.** Items recommended by at least 50% of all guidelines or 50% of guidelines with a specified systematic development process were considered universal components of studies on predictive clinical AI models.

include in this systematic review. Second, our systematic review included only a basic assessment of the risk of bias, differentiating between expert-led, collaborative and comprehensive guidelines by analyzing the rigor of the consensus process. While risk of bias assessment tools developed for systematic reviews of observational or interventional trials[24,25] would not be appropriate for a methodological review, an in-depth analysis with a custom, methods-centered tool[26] could have provided more insights on the specific shortcomings of the included guidelines. Third, we acknowledge the potential limitation of the context-agnostic nature of our summary of consensus items. While we intentionally adopted a general-ized approach to create broadly applicable findings, we recognize that this lack of nuance may result in our findings being of varying applicability depending on the specific subject domain. Fourth, this systematic review has limitations related to guideline selection and classification and limited generalizability. To allow for focused comparison of guideline content, only those reporting guidelines offering actionable items were included. Three high-quality reporting guidelines were excluded given that they do not specifically address AI in medicine: STARD[27], STROBE[28], and SPIRIT[29,30]. While these guidelines are clearly out of the scope of this systematic review and some of these guidelines have dedicated AI-specific guidelines in development (e.g. STARD-AI), indicating that the creators of the guidelines themselves may have seen deficiencies regarding com-putational medical research, they could still have provided valuable insights. Similarly, some publications were considered out of scope for reviewing very specific areas of AI such as surrogate metrics[31] without demanding actionable items. In addition, future guideline updates could result in changes in the landscape of AI reporting guidelines, which this systematic review cannot represent. Nevertheless, this review contributes to the scientific landscape in two ways: First, it provides a resource for scientists as to what guideline to adhere to. Second, it highlights potential areas for improvement that policymakers, scientific institutions and journal editors can reinforce.

In conclusion, this systematic review provides a comprehensive over-view of existing guidelines for AI applications in medicine. While the guidelines reviewed vary in quality and scope, they generally provide valuable guidance for developing and evaluating AI-based models. How-ever, the lack of standardization across guidelines, particularly regarding the ethical, legal, and social implications of AI in healthcare, highlights the need for further research and collaboration in this area. Furthermore, as AI-based models become more prevalent in clinical practice, it will be essential to update guidelines regularly to reflect the latest developments in the field and ensure their continued relevance. Good scientific practice needs to be reinforced by every individual scientist and every scientific institution. It is the same with reporting guidelines. No guideline in itself can guarantee quality and reproducibility of research. A guideline only unfolds its power when interpreted by responsible scientists.

## Data availability

All included guidelines are publicly available. The list of guideline items included in published guidelines regulating medical AI research that was generated in this systematic review is published along with this work (Supplementary Table 1).

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

## Acknowledgements
F.R.K. is supported by the German Cancer Research Center (CoBot 2.0), the Joachim Herz Foundation (Add-On Fellowship for Interdisciplinary Life Science) and the German Research Foundation (Deutsche Forschungsgemeinschaft, DFG) as part of Germany's Excellence Strategy (EXC 2050/1, Project ID 390696704) within the Cluster of Excellence"Centre for Tactile Internet with Human-in-the-Loop" (CeTI) of Dresden University of Technology. Furthermore, F.R.K. receives support from the Indiana Clinical and Translational Sciences Institute funded, in part, by Grant Number UM1TR004402 from the National Institutes of Health, National Center for Advancing Translational Sciences, Clinical and Translational Sciences Award. G.P.V. is partly supported by BMBF (Federal Ministry of Education and Research) in DAAD project 57616814 (SECAI, School of Embedded Composite AI, https://secai.org/) as part of the program Konrad Zuse Schools of Excellence in Artificial Intelligence. J.N.K. is supported by the German Federal Ministry of Health (DEEP LIVER, ZMVI1-2520DAT111) and the Max-Eder-Programme of the German Cancer Aid (grant #70113864), the German Federal Ministry of Education and Research (PEARL, 01KD2104C; CAMINO, 01EO2101; SWAG, 01KD2215A; TRANSFORM LIVER, 031L0312A), the German Academic Exchange Service (SECAI, 57616814), the German Federal Joint Committee (Transplant.KI, 01VSF21048) the European Union (ODELIA, 101057091; GENIAL, 101096312) and the National Institute for Health and Care Research (NIHR, NIHR213331) Leeds Biomedical Research Centre. The views expressed are those of the author(s) and not necessarily those of the National Institutes of Health, the NHS, the NIHR or the Department of Health and Social Care.

## Author contributions
F.R.K., G.P.V. and J.N.K. conceptualized the study, developed the search strategy, conducted the review, curated, analyzed, and interpreted the data. F.R.K., G.P.V. and J.Z. prepared visualizations. D.T. and J.N.K. provided oversight, mentorship, and funding. F.R.K. and G.P.V. wrote the original draft of the manuscript. All authors reviewed and approved the final version of the manuscript.

## Funding

## Competing interests
D.T. holds shares in StratifAI GmbH and has received honoraria for lectures from Bayer. J.N.K. declares consulting services for Owkin, France, DoMore Diagnostics, Norway, Panakeia, UK, Scailyte, Switzerland, Cancilico, Germany, Mindpeak, Germany, MultiplexDx, Slovakia, and Histofy, UK; furthermore, he holds shares in StratifAI GmbH, Germany, has received a research grant by GSK, and has received honoraria by AstraZeneca, Bayer, Eisai, Janssen, MSD, BMS, Roche, Pfizer and Fresenius. All other authors declare no conflicts of interest.
