## [Peer Review File · Communications Medicine]

Reviewers' comments:

Reviewer #1 (Remarks to the Author):

This paper intends to provide a systematic review and meta-analysis of reporting guidelines in medical artificial intelligence. The authors started with 640 publications and included 20 for substantive review after a few steps of exclusion. The authors characterized these guidelines by the following (a) the guidelines are categorized into "comprehensive", "collaborative" or "expert-led" based on the consensus-building process. (b) the guidelines are categorized by the research phase that they were aimed at, "preclinical", "translational", "clinical". (c) The breadth of scope: "general medical AI" or "specific field of medicine". Then, the identified guidelines were analyzed with respect to their overlap in individual guideline recommendations.

The topic is very interesting. The approach of overview, characterization and analysis of the guidelines is useful. However, I do have a major concern over the selection of the 20 guidelines. I also identified some technical inaccuracies.

1. The selection of the guidelines for a substantive review and analysis is important to ensure fairness and usefulness. In the Study Selection section, the authors provided some information on the selection criteria, "Duplicate studies were removed. All search results were screened by two physicians with experience in clinical AI research (FRK and GPV) using Rayyan [11]. Subsequently, publications were excluded on the basis of: (1) not providing actionable reporting guidance or (2) reporting the intention to develop a new, as yet unpublished guideline rather than the guideline itself." IN THE rESULTS, The authors stated, "Out of these, 589 records were excluded based on Title and Abstract. Of the remaining 43 full-text articles assessed for eligibility, 23 records were excluded and 20 reporting guidelines were included..." As such, it's unclear if/how the selection process actually followed the pre-defined criteria or how it was actually done. My concern is not only because of these vague descriptions but also because of the exclusion of some guidelines that I don't see violation of the author-defined criteria or with any reasonable explanation/justification. Here are some examples (it may not be an exhaustive list as I'm not doing the comprehensive review):

(a) The guidelines included in the Discussion section, PROBAST [40,41], STARD [42], STROBE [43], and SPIRIT [44,45]. The authors stated the reason for exclusion is "that they do not specifically address AI in medicine." First of all, this criterion is not stated in the Study Selection section. Second, I disagree with this statement. These guidelines are published in Radiology, Annals of Internal Medicine, BMJ, how could they do not address AI in medicine?

(b) I happen to know the following guideline but I don't see how it has failed the authors' selection criteria:

El Naqa I, Boone JM, Benedict SH, Goodsitt MM, Chan HP, Drukker K, Hadjiiski L, Ruan D, Sahiner B., "AI in medical physics: guidelines for publication," Med Phys. 2021 Sep;48(9):4711-4714. doi: 10.1002/mp.15170. PMID: 34545957.

I suggest the authors provide more details on the selection criteria and process (even in a supplemental document if necessary). This would at least include details on every box labeled as "Records excluded" in figure 1. I suggest the authors specifically address the aforementioned excluded guidelines by either including them or providing fair and convincing justifications for not doing so.

2. The first paragraph of the Search Results section (page 8, line 138) indicates a total of $622 + 2 + 1 + 15 = 640$ records to begin with. However, the top two boxes in Figure 1 indicates a total of $632 + 10 = 642$ records. Not only the total number is different, but also the numbers of the subgroups are different too. I suggest the authors double-check and correct these.

3. The summary of Consensus in guideline items (line 196 page 8) "summarizes 214 items that were recommended by at least 50% of all guidelines or 50% of guidelines with a specified systematic development process". I feel that the usefulness of this is quite limited because it lacks context (e.g., data type, clinical trial vs. device development, ...). It would be more useful to summarize such consensus items in some meaningful categories respectively, for example, imaging-data related guidelines, clinical trial guidelines, etc.)

4. I was initially confused by the references for CONSORT-AI [21, 22] and SPIRIT-AI [21, 25] in Table 1. Then I realized that this was partly because the authors of CONSORT-AI and SPIRIT-AI published the same papers on two journals (for whatever reasons): Nature Medicine and Lancet Digit Health (see full citation below). The authors of this review chose to cite one from each journal: [22] and [25]. However, reference [22] is incomplete so please revise it. Also, I don't think reference [21] is appropriate in the way it's cited because it's a short commentary discussing the relevance of these two guidelines to pathology but it's not the guidelines themselves (as it may be misunderstood to be in the way it's cited in Table 1).

Cruz Rivera S, Liu X, Chan AW, et al. Guidelines for clinical trial protocols for interventions involving artificial intelligence: The SPIRIT-AI extension. *Nat Med* 2020; 26: 1351–1363.

Cruz Rivera S, et al. Guidelines for clinical trial protocols for interventions involving artificial intelligence: the SPIRIT-AI extension. *Lancet Digit Health* 2020;2:e549–60.

Liu X, Cruz Rivera S, Moher D, et al. Reporting guidelines for clinical trial reports for interventions involving artificial intelligence: The CONSORT-AI extension. *Nat Med* 2020; 26: 1364–1374.

Liu X, et al. Reporting guidelines for clinical trial reports for interventions involving artificial intelligence: the CONSORT-AI extension. *Lancet Digit Health* 2020;2:e537–48.

Reviewer #2 (Remarks to the Author):

Brief summary of the manuscript

This is a systematic review of reporting guidelines for AI-related medical research. The study aims to provide an overview of available reporting guidelines for researchers working in healthcare AI. It also examines the methods used to develop each included guideline. The review identifies common reporting items included across those guidelines with a systematic development process.

Overall impression of the work

This review highlights available guidelines to researchers working in AI, aiming to improve study reporting standards. Greater awareness of reporting guidelines is needed to improve the reporting of AI research. This study will help to raise awareness of the importance of these resources and potentially encourage authors to use them. It could help to inform the development of future AI-specific guidelines by identifying items commonly included across guidelines and also in the scenario where there is currently no relevant guideline available for a given study type. The only concern I would raise is around the timing of the protocol registration (as detailed below), but this is at the discretion of the editor. The review addresses an important area within healthcare AI research and I am not aware of another systematic review that has addressed this question as yet.

Specific comments

1. The study protocol appears to have a registration date from after completion of the majority of the work (25th August 2023) <https://doi.org/10.17605/OSF.IO/YZE6J>. It is usual for the protocol to be registered at the start of the systematic review (see PRISMA-P statement 2015), but this is at the discretion of the journal as to whether this is acceptable.
2. Line 2 – the PRISMA guidelines state that a systematic review should be identified as such in the title. I suggest changing “structured review” in the title to “systematic review” to comply with this.
3. Line 32 in the abstract states reporting guidelines were included up to 2023 but the methods state the search was conducted in June 2022 (line 99). Please clarify the timings of finalised searches.
4. Lines 59-70 in the introduction – the manuscript does not clearly describe what a reporting guideline is and what its aims are (e.g. guidelines to ensure inclusion of minimum essential information within a study, as per Simera et al. The EQUATOR Network and reporting guidelines: Helping to achieve high standards in reporting health research studies. *Maturitas*. 2009). This should be made clearer for readers who are less familiar with these guidelines and their purpose.
5. Lines 61-62 in the introduction state there is “no universal standard approach for the development of reporting guidelines”. Whilst this may be true, the paper could acknowledge that there is published

guidance for developers of reporting guidelines from the EQUATOR Network in: Moher et al. Guidance for developers of health research reporting guidelines. PLoS medicine. 2010.

6. Lines 112-113 in “study selection” – please state whether or not the two physicians screened search results independently, as this is not clear.

7. Figure 4b – the description of what this figure shows could be clearer. Please add further detail to clarify for the reader.

8. Discussion: The discussion does not cite other reviews of medical AI reporting guidelines as related work (see below). Whilst these are not systematic reviews, they are relevant to the discussion, so I suggest acknowledging the other work that has been done in this area:

a. Ibrahim et al. Reporting guidelines for artificial intelligence in healthcare research. Clinical & experimental ophthalmology. 2021 Jul;49(5):470-6.

b. Shelmerdine et al. Review of study reporting guidelines for clinical studies using artificial intelligence in healthcare. BMJ Health & Care Informatics. 2021;28(1).

9. Lines 261-262 in the discussion – The QUADAS and PROBAST tools are quality assessment tools for risk of bias and applicability concerns within studies. Therefore, QUADAS-AI and PROBAST-AI will not be reporting guidelines. However, these tools are being developed alongside the reporting guideline efforts and so this should be clarified within this section.

10. Line 296 – As above, PROBAST is a risk of bias and applicability tool, rather than a reporting guideline (Wolff et al. PROBAST: a tool to assess the risk of bias and applicability of prediction model studies. Annals of internal medicine. 2019). This should also be clarified here.

Reviewer #3 (Remarks to the Author):

David Moher

Precis

The authors report conducting a systematic review of medical artificial intelligence reporting guidelines. They included 20 reporting guidelines published between 2009 and 2023. The authors analyzed the contents of the reporting guidelines for a variety of characteristics. Some

of the graphics in the results section are excellent and highly informative for readers.

Assessment

This is a very fast-moving field (AI reporting guidelines) and the authors are to be congratulated for completing the review. Below, I've provided some suggestions and questions of clarification.

I'm hoping my suggestions will improve the usefulness of the research product for readers.

1. My main concern is the lack of any risk of bias assessment reported for this 'methods' systematic review. While I appreciate it is inappropriate to use existing risk of bias tools (developed for intervention/observational reviews), I think it is important for methods reviews to include some assessment of risk of bias. This paper provides an example of how to think about this issue and implement a tool for a methods review (Cukier S, Checklists to detect potential predatory biomedical journals: a systematic review. *BMC Med.* 2020 May 7;18(1):104. doi: 10.1186/s12916-020-01566-1). Given the importance of risk of bias assessments in any systematic review, this issue needs resolution prior to any possible publication.

2. The title should be modified to delete "structured" and replace it with "systematic".

3. On line 72, edit "...more medical AI is the scope thereof" to ... "more medical AI is its scope".

4. On line 85, please delete "comprehensive". By definition, a systematic review is comprehensive.

5. On line 98, the authors refer to using PRISMA to guide the conduct of their systematic review. PRISMA is a reporting guideline and not a methods guide. This should be modified.

6. For the search strategy section, who developed and executed the search strategies? Was a librarian involved. Clarification would be helpful.

a. Similarly, did the authors search the EQUATOR Network's library of reporting guidelines (<https://www.equator-network.org/library/>),

b. and were any preprint servers searched? Clarification in the paper would be helpful. Finally, was any search strategy PRESSed (peer review of electronic search strategies) (McGowan J. PRESS Peer Review of Electronic Search

Strategies: 2015 Guideline Statement. *J Clin Epidemiol.* 2016 Jul;75:40-6. doi:

10.1016/j.jclinepi.2016.01.021)?

7. For the study selection section, do the authors have any agreement data to report/share with readers?

a. Similarly, how were disagreements resolved?

8. For the data extraction and analysis section, the authors could have used a tool to (Schlüssel MM et al. Reporting guidelines used varying methodology to develop recommendations. J Clin Epidemiol. 2023 Jul;159:246-256. doi:

10.1016/j.jclinepi.2023.03.018. Epub 2023 Mar 24. PMID: 36965598 or Moher D.

Guidance for developers of health research reporting guidelines. PLoS Med. 2010 Feb 16;7(2):e1000217. doi: 10.1371/journal.pmed.1000217. PMID: 20169112; PMCID: PMC2821895.). Please clarify why none of these tools were considered/used.

9. Please describe the data extraction from in some detail for readers.

10. On line 154, immediately following the landscape of reporting guidelines in clinical AI, delete “We included a total of 20 reporting guidelines in the present review”. This is redundant here and was previously reported earlier on in the results.

11. Do any of the 20 included guidelines mention/include anything about open science? Any mention of data and/or code sharing? Please report on both points.

12. For the discussion section, I would delete the first three paragraphs and replace them with a single paragraph, providing a summary of the results. I would also recommend a strengths/limitations section for the discussion.

13. On line 281, I was unsure what the authors meant by “...some use cases remain poorly regulated.”. Please clarify.

Point by Point Response

Reviewer #1:

This paper intends to provide a systematic review and meta-analysis of reporting guidelines in medical artificial intelligence. The authors started with 640 publications and included 20 for substantive review after a few steps of exclusion. The authors characterized these guidelines by the following (a) the guidelines are categorized into "comprehensive", "collaborative" or "expert-led" based on the consensus-building process. (b) the guidelines are categorized by the research phase that they were aimed at, "preclinical", "translational", "clinical". (c) The breadth of scope: "general medical AI" or "specific field of medicine". Then, the identified guidelines were analyzed with respect to their overlap in individual guideline recommendations.

The topic is very interesting. The approach of overview, characterization and analysis of the guidelines is useful. However, I do have a major concern over the selection of the 20 guidelines. I also identified some technical inaccuracies.

Thank you very much for your thorough review of our manuscript. Based on your suggestions and comments, we have subjected our work to major changes, which we feel have considerably improved the clarity and reproducibility of our work. Please find our detailed responses below.

1. The selection of the guidelines for a substantive review and analysis is important to ensure fairness and usefulness. In the Study Selection section, the authors provided some information on the selection criteria, "Duplicate studies were removed. All search results were screened by two physicians with experience in clinical AI research (FRK and GPV) using Rayyan [11]. Subsequently, publications were excluded on the basis of: (1) not providing actionable reporting guidance or (2) reporting the intention to develop a new, as yet unpublished guideline rather than the guideline itself." In the results, The authors stated, "Out of these, 589 records were excluded based on Title and Abstract. Of the remaining 43 full-text articles assessed for eligibility, 23 records were excluded and 20 reporting guidelines were included..." As such, it's unclear if/how the selection process actually followed the pre-defined criteria or how it was actually done. My concern is not only because of these vague descriptions but also because of the exclusion of some guidelines that I don't see violation of the author-defined criteria or with any reasonable explanation/justification. Here are some examples (it may not be an exhaustive list as I'm not doing the comprehensive review):

Thank you very much for bringing this point to our attention. We agree that our description of the inclusion and exclusion process required more elaboration. Therefore, we have added details on the search strategy as well as inclusion and exclusion criteria to the Methods section of our manuscript:

"... The inclusion criteria were (1) the topic of the publication being AI in medicine and (2) the guideline recommendations being specific to the application of AI methods for either preclinical, translational, or clinical scenarios. Publications were excluded on the basis of (1) not providing actionable reporting guidance, (2) collecting or reassembling guideline items from existing guidelines rather than providing new guideline items or (3)

reporting the intention to develop a new, as yet unpublished guideline rather than the guideline itself. ...”

(a) The guidelines included in the Discussion section, PROBAST [40,41], STARD [42], STROBE [43], and SPIRIT [44,45]. The authors stated the reason for exclusion is "that they do not specifically address AI in medicine." First of all, this criterion is not stated in the Study Selection section. Second, I disagree with this statement. These guidelines are published in Radiology, Annals of Internal Medicine, BMJ, how could they do not address AI in medicine?

Thank you very much. As outlined above, the revised version of our manuscript now includes more details on the inclusion and exclusion criteria, including the thematic focus on “AI in medicine”. The PROBAST, STARD, STROBE and SPIRIT guidelines were excluded since our review focuses on guidelines that are specific to AI-based studies. While we agree that these guidelines are of very high quality and importance to medical research, they do not address peculiarities of computational research (such as the selection and preprocessing of data, the selection of computational models, or validation (i.e., presence of a hold-out test set), and were therefore excluded. Moreover, some of these guidelines have dedicated AI-specific guidelines in development (e.g. STARD-AI), indicating that the creators of the guidelines themselves may have seen them as deficient for AI-related research. We have added this explanation to the discussion section of the revised manuscript.

(b) I happen to know the following guideline but I don't see how it has failed the authors' selection criteria:

El Naqa I, Boone JM, Benedict SH, Goodsitt MM, Chan HP, Drukker K, Hadjiiski L, Ruan D, Sahiner B., “AI in medical physics: guidelines for publication,” Med Phys. 2021 Sep;48(9):4711-4714. doi: 10.1002/mp.15170. PMID: 34545957.

We appreciate that you have brought this guideline to our attention. This guideline was not part of the results of our initial search. Given its relevance to the topic of this systematic review, we have included it as an incidental finding and have added it throughout the manuscript.

I suggest the authors provide more details on the selection criteria and process (even in a supplemental document if necessary). This would at least include details on every box labeled as "Records excluded" in figure 1. I suggest the authors specifically address the aforementioned excluded guidelines by either including them or providing fair and convincing justifications for not doing so.

Thank you very much. As pointed out above, we have included details on the study selection process and have included the guideline published by El Naqa et al. as an incidental finding.

2. The first paragraph of the Search Results section (page 8, line 138) indicates a total of $622 + 2 + 1 + 15 = 640$ records to begin with. However, the top two boxes in Figure 1 indicate a total of $632 + 10 = 642$ records. Not only the total number is different, but also the numbers of the subgroups are different too. I suggest the authors double-check and correct these.

We appreciate the reviewer's diligence in pointing out the discrepancies in the reported numbers in our manuscript. We have thoroughly reevaluated our records and have rectified the inconsistencies. Factoring adjustments based upon reviewer feedback, we have corrected the numbers in the Search Results section to ensure they are accurate and in alignment with our study's findings. We have also ensured that the numbers for the subgroups are consistent with these revised totals. Thank you for bringing this to our attention, and we apologize for any confusion it may have caused.

3. The summary of Consensus in guideline items (line 196 page 8) "summarizes 214 items that were recommended by at least 50% of all guidelines or 50% of guidelines with a specified systematic development process". I feel that the usefulness of this is quite limited because it lacks context (e.g., data type, clinical trial vs. device development, ...). It would be more useful to summarize such consensus items in some meaningful categories respectively, for example, imaging-data related guidelines, clinical trial guidelines, etc.)

We empathize with your concern regarding the perceived limitations of our summary, stemming from a lack of contextual details. We acknowledge the significance of offering more detailed categorization to improve the relevance of our findings to specific subject domains. Despite this, we want to emphasize that our choice to maintain a level of generality was intentional, aiming to create broadly applicable findings for a diverse audience within the scope of guideline development. We recognize the trade-offs associated with this approach and value your input. In response to your concern, we have endeavored to enhance transparency by noting this in the "limitations" paragraph of our article. These additions aim to explicitly address the context-agnostic nature of our stance, acknowledging potential limitations for specific subject domains. We believe this adjustment aligns with your expectations and enhances understanding of our study's scope and potential constraints.

4. I was initially confused by the references for CONSORT-AI [21, 22] and SPIRIT-AI [21, 25] in Table 1. Then I realized that this was partly because the authors of CONSORT-AI and SPIRIT-AI published the same papers on two journals (for whatever reasons): Nature Medicine and Lancet Digit Health (see full citation below). The authors of this review chose to cite one from each journal: [22] and [25]. However, reference [22] is incomplete so please revise it. Also, I don't think reference [21] is appropriate in the way it's cited because it's a short commentary discussing the relevance of these two guidelines to pathology but it's not the guidelines themselves (as it may be misunderstood to be in the way it's cited in Table 1).

Cruz Rivera S, Liu X, Chan AW, et al. Guidelines for clinical trial protocols for interventions involving artificial intelligence: The SPIRIT-AI extension. *Nat Med* 2020; 26: 1351–1363.

Cruz Rivera S, et al. Guidelines for clinical trial protocols for interventions involving artificial intelligence: the SPIRIT-AI extension. *Lancet Digit Health* 2020;2:e549–60.

Liu X, Cruz Rivera S, Moher D, et al. Reporting guidelines for clinical trial reports for interventions involving artificial intelligence: The CONSORT-AI extension. *Nat Med* 2020; 26: 1364–1374.

Liu X, et al. Reporting guidelines for clinical trial reports for interventions involving artificial intelligence: the CONSORT-AI extension. *Lancet Digit Health* 2020;2:e537–48.

Thank you for your feedback. We appreciate your observation and have addressed the concerns regarding the references to CONSORT-AI and SPIRIT-AI in Table 1. We have

revised reference [22] to ensure its completeness, and we recognize that reference [21] could be misleading in the way it was cited. We have reviewed the citation and made the necessary adjustments, namely removing reference [21] and [22] and providing a new reference for CONSORT-AI.

Reviewer #2:

Brief summary of the manuscript

This is a systematic review of reporting guidelines for AI-related medical research. The study aims to provide an overview of available reporting guidelines for researchers working in healthcare AI. It also examines the methods used to develop each included guideline. The review identifies common reporting items included across those guidelines with a systematic development process.

Overall impression of the work

This review highlights available guidelines to researchers working in AI, aiming to improve study reporting standards. Greater awareness of reporting guidelines is needed to improve the reporting of AI research. This study will help to raise awareness of the importance of these resources and potentially encourage authors to use them. It could help to inform the development of future AI-specific guidelines by identifying items commonly included across guidelines and also in the scenario where there is currently no relevant guideline available for a given study type. The only concern I would raise is around the timing of the protocol registration (as detailed below), but this is at the discretion of the editor. The review addresses an important area within healthcare AI research and I am not aware of another systematic review that has addressed this question as yet.

Thank you very much for your appreciation of our work. We agree that our study can help to raise awareness for the importance of these guidelines and to foster their use. We have replied to your concerns point-by-point below.

Specific comments

1. The study protocol appears to have a registration date from after completion of the majority of the work (25th August 2023) <https://doi.org/10.17605/OSF.IO/YZE6J>. It is usual for the protocol to be registered at the start of the systematic review (see PRISMA-P statement 2015), but this is at the discretion of the journal as to whether this is acceptable.

Thank you very much for bringing up this important point, which we have discussed with the editorial team at Communications Medicine during the submission process. Since the PROSPERO database, the most commonly used database of systematic reviews, only considers systematic reviews on health-related outcomes and we were not aware of other databases for systematic reviews on “endpoints” other than health-related outcomes, we had not prospectively registered this trial. We have discussed the option of retrospective registration with the editorial board who agreed that this procedure is acceptable. Overall, while we indeed opted for a retrospective registration process, we would like to stress that this did not change the set up or the course of our study.

2. Line 2 – the PRISMA guidelines state that a systematic review should be identified as such in the title. I suggest changing “structured review” in the title to “systematic review” to comply with this.

Thank you very much for bringing up this point, which was also raised by Reviewer #3. We have replaced “structured” with “systematic” in the manuscript title.

3. Line 32 in the abstract states reporting guidelines were included up to 2023 but the methods state the search was conducted in June 2022 (line 99). Please clarify the timings of finalised searches.

Thank you for bringing this to our attention. We conducted the PubMed search in June 2022, but added additional incidentally found studies during the process of writing and the review process. Therefore, our review contains studies that were published in 2023. To guide the attentive reader, the statement in the methods section now reads:

“...Clarivate Journal Citation reports were screened for specific AI/ML checklist requirements for submitted articles. Studies identified as incidental findings were added independent of the aforementioned search process, thereby including studies published after June 26, 2022.”

4. Lines 59-70 in the introduction – the manuscript does not clearly describe what a reporting guideline is and what its aims are (e.g. guidelines to ensure inclusion of minimum essential information within a study, as per Simera et al. The EQUATOR Network and reporting guidelines: Helping to achieve high standards in reporting health research studies. Maturitas. 2009). This should be made clearer for readers who are less familiar with these guidelines and their purpose.

We appreciate the reviewer's feedback and have carefully revised the second paragraph of the introduction to better address these concerns. We have incorporated a more explicit definition of reporting guidelines and their objectives, drawing upon the reference by Simera et al. (2009) to elucidate the purpose of these guidelines: to ensure the inclusion of minimum essential information within research studies. We believe that these adjustments provide a more comprehensive understanding of reporting guidelines for all readers, including those who may be less familiar with the subject.

5. Lines 61-62 in the introduction state there is “no universal standard approach for the development of reporting guidelines”. Whilst this may be true, the paper could acknowledge that there is published guidance for developers of reporting guidelines from the EQUATOR Network in: Moher et al. Guidance for developers of health research reporting guidelines. PLoS medicine. 2010.

Thank you very much for bringing this point to our attention, which has also been mentioned by Reviewer #3. We have accordingly integrated existing recommendations on guideline development into the discussion section of the manuscript.

6. Lines 112-113 in “study selection” – please state whether or not the two physicians screened search results independently, as this is not clear.

Thank you very much. The two physicians screened search results independently; screening results were blinded until completion of their screening. Subsequently, disagreements regarding the inclusion/exclusion of a guideline were resolved through judgment of a third reviewer (JNK). The same approach was applied to individual guideline items. These details have been added to the methods section of the revised

manuscript. Moreover, as suggested by Reviewer #1, we now outline the selection criteria in more detail in the Methods section of the manuscript.

7. Figure 4b – the description of what this figure shows could be clearer. Please add further detail to clarify for the reader.

We agree and have revised both figure 4b itself (better legibility of labels inside figure) and the corresponding figure text to better explain what this figure shows.

8. Discussion: The discussion does not cite other reviews of medical AI reporting guidelines as related work (see below). Whilst these are not systematic reviews, they are relevant to the discussion, so I suggest acknowledging the other work that has been done in this area:

- a. Ibrahim et al. Reporting guidelines for artificial intelligence in healthcare research. *Clinical & experimental ophthalmology*. 2021 Jul;49(5):470-6.
- b. Shelmerdine et al. Review of study reporting guidelines for clinical studies using artificial intelligence in healthcare. *BMJ Health & Care Informatics*. 2021;28(1).

We appreciate these recommendations and have accordingly adapted the Discussion section of our manuscript. The revised manuscript now contains a brief outline of related works and the differentiation of our review from existing literature.

9. Lines 261-262 in the discussion – The QUADAS and PROBAST tools are quality assessment tools for risk of bias and applicability concerns within studies. Therefore, QUADAS-AI and PROBAST-AI will not be reporting guidelines. However, these tools are being developed alongside the reporting guideline efforts and so this should be clarified within this section.

Thank you. We have clarified the discussion section in terms of the QUADAS-AI and PROBAST-AI quality assessment tools. Thank you very much for pointing out this inaccuracy.

10. Line 296 – As above, PROBAST is a risk of bias and applicability tool, rather than a reporting guideline (Wolff et al. PROBAST: a tool to assess the risk of bias and applicability of prediction model studies. *Annals of internal medicine*. 2019). This should also be clarified here.

We agree and have revised this in the discussion. Once again, thank you very much for your thoughtful comments and for sharing your expertise. We believe that your input has substantially improved the clarity and accuracy of our work.

Reviewer #3:

Precis

The authors report conducting a systematic review of medical artificial intelligence reporting guidelines. They included 20 reporting guidelines published between 2009 and 2023. The authors analyzed the contents of the reporting guidelines for a variety of characteristics. Some of the graphics in the results section are excellent and highly informative for readers.

Assessment

This is a very fast-moving field (AI reporting guidelines) and the authors are to be congratulated for completing the review. Below, I've provided some suggestions and questions of clarification. I'm hoping my suggestions will improve the usefulness of the research product for readers.

We would like to thank Reviewer #3 for their detailed review of our manuscript and for helpful suggestions and comments, some of which were also mentioned by Reviewers #1 and #2. We have adapted our work accordingly and feel that the changes have considerably improved our work in terms of reproducibility and clarity.

1. My main concern is the lack of any risk of bias assessment reported for this 'methods' systematic review. While I appreciate it is inappropriate to use existing risk of bias tools (developed for intervention/observational reviews), I think it is important for methods reviews to include some assessment of risk of bias. This paper provides an example of how to think about this issue and implement a tool for a methods review (Cukier S, Checklists to detect potential predatory biomedical journals: a systematic review. *BMC Med.* 2020 May 7;18(1):104. doi: 10.1186/s12916-020-01566-1). Given the importance of risk of bias assessments in any systematic review, this issue needs resolution prior to any possible publication.

Thank you very much for bringing up this important point. Assessing the quality of the landscape of AI-related reporting guidelines and identifying the frequency of shortcomings in guideline creation was one of the major goals of this systematic review. We would like to point out that we used a criterion to assess the risk of bias in each reporting guideline, focusing on the reported strategy to construct the guideline. The results of this assessment are summarized in Table 1 and visualized in Figures 3 and 4 ("Consensus Process"). The consensus process of a guideline was classified as "expert-led" if there was no formal procedure for establishing the guideline, "collaborative" if the authors (presumably) used a formal consensus procedure involving multiple experts, but provided no details on the exact protocol. Guidelines classified as "comprehensive" outlined a clear formal consensus procedure involving multiple experts with details on the exact protocol.

We have carefully reviewed the resource that you provided in your comment and acknowledge that we could have performed a more in-depth review of the risk of bias. However, we would like to keep the current basic level of a risk of bias assessment for two main reasons: First, we see substantial overlap between the criterion "Represent 1+ stakeholder groups" used in the work by Cukier et al. (2020) and our classification of the

consensus process outlined above. Second, we feel that some of the additional criteria used in the work by Cukier et al. (i.e. “fits on one page”, “each item one sentence” are not specifically relevant to the quality of AI-related reporting guidelines.

We have slightly rephrased the wording throughout the manuscript to clarify that the assessment of the consensus process is a basic form of a risk of bias assessment. In addition, we have added the single-criterion character of our risk of bias assessment as a limitation of our work, outlining that a more detailed risk of bias assessment could have provided more insights on the specific shortcomings of each included guideline. We hope that these changes satisfactorily address the risk of bias assessment.

2. The title should be modified to delete “structured” and replace it with “systematic”.

Thank you very much for bringing this to our attention. We have replaced “structured” with “systematic” in the manuscript title.

3. On line 72, edit “...more medical AI is the scope thereof” to ... “more medical AI is its scope”.

Thank you for this suggestion. We have changed the manuscript accordingly.

4. On line 85, please delete “comprehensive”. By definition, a systematic review is comprehensive.

Thank you. We agree and have deleted “comprehensive”.

5. On line 98, the authors refer to using PRISMA to guide the conduct of their systematic review. PRISMA is a reporting guideline and not a methods guide. This should be modified.

Thank you very much for pointing out this inaccuracy. We have accordingly modified this sentence.

6. For the search strategy section, who developed and executed the search strategies? Was a librarian involved. Clarification would be helpful.

Thank you very much. The authors (FRK, GPV, JNK) developed the search strategy. No librarian was involved. We have added this as a limitation of our work.

- a. Similarly, did the authors search the EQUATOR Network’s library of reporting guidelines (<https://www.equator-network.org/library/>),

Thank you very much for bringing this tool to our attention. Our initial search did not include a search of the EQUATOR Network’s library of reporting guidelines. To ensure completeness of our systematic review, we have conducted a post-hoc search of the EQUATOR Network’s library of reporting guidelines. This search identified 8 additional guidelines that we have added to the review, out of which 5 were ultimately included in the selected studies. We have accordingly updated the Methods, Results, and Discussion section of our manuscript to display these changes. We very much appreciate your input on this.

b. and were any preprint servers searched? Clarification in the paper would be helpful. Finally, was any search strategy PRESSed (peer review of electronic search strategies) (McGowan J. PRESS Peer Review of Electronic Search Strategies: 2015 Guideline Statement. J Clin Epidemiol. 2016 Jul;75:40-6. doi: 10.1016/j.jclinepi.2016.01.021)?

No preprint servers were searched and the search strategy was not PRESSed. We now clarify this in the Methods section and have added the limitations related to the search strategy to the limitations section of the manuscript.

7. For the study selection section, do the authors have any agreement data to report/share with readers?

a. Similarly, how were disagreements resolved?

Thank you. As mentioned in our manuscript, two independent reviewers (FRK and GPV) screened all search results. Disagreements were resolved by judgment of a third reviewer (JNK). Agreement data regarding study selection are outlined at the end of the first section of the Results:

“Interrater agreement for study selection on the basis of full-text records was 71% (n = 15 requiring third reviewer out of n = 52).”

8. For the data extraction and analysis section, the authors could have used a tool to (Schlüssel MM et al. Reporting guidelines used varying methodology to develop recommendations. J Clin Epidemiol. 2023 Jul;159:246-256. doi: 10.1016/j.jclinepi.2023.03.018. Epub 2023 Mar 24. PMID: 36965598 or Moher D. Guidance for developers of health research reporting guidelines. PLoS Med. 2010 Feb 16;7(2):e1000217. doi: 10.1371/journal.pmed.1000217. PMID: 20169112; PMCID: PMC2821895.). Please clarify why none of these tools were considered/used.

Thank you very much for pointing this out and for bringing these tools to our attention. While we agree that these tools are relevant for the development of new guidelines, we feel that they are not completely transferable to our case of analyzing and comparing existing guidelines. Overall, we considered the blinded, manual review by two researchers using the Rayyan platform, with arbitration by a third researcher in case of discrepancy, a valid ground truth for our problem. However, we now briefly describe some development recommendations for new guidelines in the Discussion and have added the respective references.

9. Please describe the data extraction from in some detail for readers.

Thank you very much. We agree that the data extraction methods required clarification and have added respective details to the Methods section of the manuscript.

10. On line 154, immediately following the landscape of reporting guidelines in clinical AI, delete “We included a total of 20 reporting guidelines in the present review”. This is redundant here and was previously reported earlier on in the results.

Thank you. We agree and have deleted this sentence.

11. Do any of the 20 included guidelines mention/include anything about open science? Any mention of data and/or code sharing? Please report on both points.

Thank you very much. Data and/or code publication were both criteria that were reviewed when comparing the concordance of guideline items (Figure 4, Figure 5, Supplementary Table 1). Code publication is mentioned by 15 of the 26 guidelines (6/9 comprehensive guidelines), while data publication is part of 12 of the 26 guidelines (4/9 comprehensive guidelines).

12. For the discussion section, I would delete the first three paragraphs and replace them with a single paragraph, providing a summary of the results. I would also recommend a strengths/limitations section for the discussion. 13. On line 281, I was unsure what the authors meant by "...some use cases remain poorly regulated.". Please clarify.

Thank you very much for these suggestions. We agree and have changed the Discussion accordingly, including a summary of the results as well as expanding the discussion of the strengths and limitations of our work, also taking some suggestions by Reviewer #2 into account. We hope that these changes satisfactorily address your concerns and once again thank you for the thorough review of our work and your helpful recommendations.

REVIEWERS' COMMENTS:

Reviewer #1 (Remarks to the Author):

My comments have been addressed satisfactorily.

Reviewer #2 (Remarks to the Author):

Thank you to the authors for addressing all of my comments on the manuscript. I am satisfied that all of my points have been addressed.

Reviewer #3 (Remarks to the Author):

I have previously reviewed this paper. I reviewed all three peer reviewers comments against the modifications the authors made to the paper. I congratulate the authors. The paper is much improved and IMO will be very useful to the broad readership of the journal.

Point by Point Response

Reviewer #1:

My comments have been addressed satisfactorily.

Thank you very much for your input and comments.

Reviewer #2:

Thank you to the authors for addressing all of my comments on the manuscript. I am satisfied that all of my points have been addressed.

We appreciate your comments and input, which have helped improve our paper.

Reviewer #3:

I have previously reviewed this paper. I reviewed all three peer reviewers comments against the modifications the authors made to the paper. I congratulate the authors. The paper is much improved and IMO will be very useful to the broad readership of the journal.

We appreciate your thorough re-review of our work and for spotting minor issues, which we have corrected based on your provided PDF. Your input has considerably improved our work.